# Online Instruction to Measure Axial Alignment with the Bonesetter App

**DOI:** 10.3390/medicina58081079

**Published:** 2022-08-10

**Authors:** Mitchell Bernstein, Tatiana Bunge, Kadence Rosinski, Mauricio Kfuri, Brett Crist, Andrew Knapp, Rahul Vaidya

**Affiliations:** 1Department of Orthopedic Surgery, McGill University, Montreal, QC H3A 0G4, Canada; 2Department of Orthopedic Surgery, Detroit Medical Center, Detroit, MI 48124, USA; 3Department of Orthopedic Surgery, University of Missouri, Columbia, MO 65211, USA

**Keywords:** axial alignment, femoral version, tibial torsion, Bonesetter app, retroversion, anteversion, deformity correction

## Abstract

*Background and objectives*: Alignment of the lower extremity is important when treating congenital deformities, fractures, and joint replacement. During the COVID-19 pandemic, AO North America offered an online course on deformity measurement and planning. The Bonesetter app is a deformity planning tool that is freely available online. The purpose of this study was to see how effective an online course was in teaching axial alignment measurement and to assess that skill using an online digital planning tool, the Bonesetter app. *Materials and Methods*: An online module on axial alignment was provided during the AONA osteotomy course as well as a tutorial on how to use an online digital planning tool (Bonesetter app). The tools within the Bonesetter app allow users to draw digital lines directly on the CT images and measure the exact angle between two planes. Participants in this study were directed to perform these measurements in four different cases that tested different variations of deformity. *Results*: The measurements were completed correctly in case 1 = 56%, case 2 = 61%, case 3 = 84%, and case 4 = 76%. The standard deviation of angular digital measurements between individuals was ±4.26 degrees. Measuring the angle directly vs. drawing angles to a horizontal line had smaller standard deviations per case (*p* < 0.005) and less incidents outside 1 standard deviation for each measurement. Errors in adding and subtracting were the most common errors, particularly in relation to femoral anteversion or retroversion. *Conclusions*: The online course successfully instructed a group of orthopedic surgeons to measure alignment and malalignment of lower limb axial deformities. The Bonesetter app helped participants to learn this skill and identify errors in measurement. The inability to differentiate between anteversion and retroversion of the femur is a common source of error when it occurs and should be a focus of instruction.

## 1. Introduction

Defining a consistent method of measuring both femoral version and tibial torsion has been debated in orthopedic literature and there is currently no standard practice in place. The alignment of the lower extremity is important when treating congenital deformities, fractures, and joint replacement. Rotational deformities can cause gait dysfunction, pain, and patellofemoral instability [1]. Rotational assessment through physical examination has proven to be inaccurate, particularly in the femur [2]. Computed Tomography (CT) rotation studies are used in addition to clinical observation to provide an objective measure of the torsional deformity and is considered the gold standard to measure version [3,4,5,6].

Femoral anteversion is the angle in the transverse plane between the axis of the femoral neck (center of the femoral head to the center of the femoral neck base) in reference to the condylar axis, which is parallel to the posterior femoral condyles [7,8]. Goutallier defines tibial torsion as the angle between the proximal tibial line (CT cut immediately distal to the knee joint) and the bimalleolar line (the first section distal to the tibial pilon in which the posterior malleolus was not visible) while others use the distal femoral condylar axis and the bimalleolar line [9]. Notably, there exists a wide variation in defined normal femoral version, between 8 and 14°, similarly, the normal range for tibial torsion is between 18.5 and 33.7° [10,11,12].

The Bonesetter app (Bonesetter Solutions LLC Ann Arbor Michigan) was developed to aid in surgical planning of deformity, templating fracture reduction, and joint replacement (https://detroitbonesetter.com/user/login, accessed on 18 April 2020) [13]. It also has an online feature which allows educators to prepare exercises, place them online, test participants, and then give them individual feedback in real time on their work (https://learn.detroitbonesetter.com/app) (accessed on 18 April 2020).

The COVID-19 pandemic caused many educational programs to be converted to an online format. AO (Arbeitsgemeinschaft für Osteosynthesefragen or Association of the Study of Internal Fixation) North America (AONA) led a ten-week osteotomy webinar to instruct a group of surgeons on deformity measurement and treatment. The module on axial alignment taught participants how to make rotational assessments of the lower extremities and measure deformities. The Bonesetter app was used to teach and then assess if participants could replicate this skill on four test cases. The purpose of this trial was to test how well orthopedic practitioners were able to measure axial alignment and malalignment using an online digital planning tool.

## 2. Materials and Methods

An IRB approved study was performed to evaluate an educational event’s ability to teach the skill of axial plane measurement in the lower extremity. During the AONA online osteotomy course, a module was prepared to teach axial alignment and malalignment of the lower extremity [2,14,15,16]. Dr. Bernstein taught a 30 min tutorial as part of a larger session on axial alignment on CT scans using the Bonesetter app. The participants were residents, fellows, and attendings from 4 different continents (North America, South America, Europe, and Asia) and all were members of the AONA. Four different cases were created online to test this skill and we evaluated their proficiency in replicating the measurements of femoral version and tibial torsion [2]. Each case provided CT version studies showing femoral neck axis, posterior condylar axis, and bimalleolar axis to practice using the angle measuring tools and analyze the degree of rotation. The Bonesetter app is unique because it allows course instructors to view data saved by participants in real time to ensure the success of their students. To measure the angle of the femoral neck axis, the participants used the angle tool to draw two lines, one connecting the centers of the femoral head and neck and another horizontal reference line (Figure 1a,b). To measure the angle at the talocrural joint, participants used the angle tool to draw a line connecting the centers of the medial and lateral malleoli and another horizontal reference line (Figure 1b,c). To measure the angle of the tibiofemoral joint, participants used the angle tool to draw one line connecting the posterior surfaces of the condyles and another horizontal reference line (Figure 1b,c and Figure 2a). Alternatively, participants could measure the angles directly, i.e., femoral neck to the posterior condylar axis of the femur and the posterior condylar axis of the femur to the bimalleolar axis (Figure 2b).

To calculate femoral version, participants found the difference between the angles of the femoral neck and posterior condyles. To calculate tibial torsion, participants found the difference between the angles of the posterior condyles and the bimalleolar axis. The participants saved their work on the app, and a feature within the app allows the instructor to view each participant’s work. In addition, participants needed to find the differences between the right and left extremities and record these in the notes section. Two independent observers who had been taught how to measure alignment on the app evaluated the work saved by the participants, which was judged for being correct or not and the cause of mistakes was analyzed. We were also able to measure standard deviation of the measurements to assess how well the measurements could be reproduced or what the expected variability of measurements was on the digital planning tool.

To test the accuracy of measurements between observers, a two-sample independent t-test was performed comparing the standard deviations of the two methods. The sample size was 32, degrees of freedom set to 30 and α (1). To compare occurrence of error, a Mann–Whitney U test was performed by analyzing the number of participants measuring outside of one standard deviation from the example (cases 1–3) or the mean (case 4). The number of cases for each method was ranked from low to high, n_1_ = 16, n_2_ = 16, α (1).

The recordings of each instructional video instructing participants can be found by clicking the following links:

Case 1: https://www.youtube.com/watch?v=HL8wq8mUL7o&list=PLK_TIY7xyD45C4upDXXY5xK28_3P4k6t&index=46&t=24s (accessed on 18 April 2020).

Case 2: https://www.youtube.com/watch?v=4y_OYq7RZlQ&list=PLK_-TIY7xyD45C4upDXXY5xK28_3P4k6t&index=46 (accessed on 18 April 2020).

Case 3: https://www.youtube.com/watch?v=5RJNvpJ5R00&list=PLK_-TIY7xyD45C4upDXXY5xK28_3P4k6t&index=45 (accessed on 18 April 2020).

Full deformity evaluation online course: https://www.youtube.com/watch?v=REFXfzoHUY0&list=PLK_-TIY7xyD45C4upDXXY5xK28_3P4k6t&index=47&t=3439s (accessed on 18 April 2020).

The free Bonesetter app is found at https://detroitbonesetter.com/user/login (accessed on 18 April 2020).

## 3. Results

In order to complete these exercises correctly, participants had to measure the three angles of the proximal femur, the distal femur, and the distal tibia correctly in each limb, determine if the femur was anteverted or retroverted, if the tibia had external or internal tibial torsion, add or subtract depending on the version or torsion of the extremity, and then correctly ascertain the difference between the two extremities. All measurements were completed correctly by only 56% of participants in case 1, 61% of participants in case 2, 84% of participants in case 3, and 76% of participants in case 4. We analyzed what errors were made by the participants, determined if it was due to learning the digital planning platform, critical errors of measurement, or calculation, and then gave feedback on how to correct their mistakes. Reproducibility of these measurements on the digital app was calculated. We then made recommendations on how teaching this skill could be improved in the future.

### 3.1. Accuracy of Course Participants after Course Instruction

#### 3.1.1. Results from Case 1 (80 Entries)

Case 1 is a simple case where the measurements are symmetrical yet only 56% of the participants who attempted this case completed it correctly. Only one participant did not place any angles (unable to use the APP) while 35% measured the angles correctly on one side but did not complete the exercise for both legs, thus they understood how to perform the exercise but just did not complete it.

#### 3.1.2. Case 2 (61 Entries)

Case 2 requires understanding that you must distinguish between anteversion and retroversion for the femur. Participants must add the angles of the left hip in anteversion and right hip in retroversion to obtain the difference in version (Figure 3). Sixty-one percent of the participants performed the exercise correctly, 30% did not complete the exercise, only measuring one side, 6.6% measured the angles incorrectly, and 1.6% calculated the numbers wrong by adding or subtracting inappropriately.

#### 3.1.3. Case 3 (55 Entries)

In case 3, 84% of the participants performed the measurements correctly and 96% completed the measurements. This case has both femurs in retroversion, thus calculating the angles again is required (Figure 4). Errors occurred in measuring the angles and calculating them incorrectly in 13% of cases.

#### 3.1.4. Case 4 (42 Data Entries)

Case 4 is a fracture case with varying amounts of femoral retroversion—left greater than right. Here, 76% of participants performed the measurements correctly and 93% of them completed the exercise. However, in 16.5% of the cases there was an error in measurement or calculation.

Overall, just as in other online teaching tools, the Bonesetter app had some attrition of users during the exercise (Table 1). Participants’ responses were included in the study based on completeness and accuracy to determine the precision and reproducibility of recording the same measurements based on instruction.

##### Accuracy of Measurements

The average standard deviation of measurements using the digital planning app for all the measurements was ±4.26 degrees. This is what we can expect variance for these measurements to be between individuals who do the exercises correctly.

##### Variance between Methods

Using a horizontal line to measure the angles vs. a direct measurement (as seen in Figure 1): the variance of measurement by using the horizontal line method (Figure 2a, 493 measurements) vs. the direct angle method (Figure 2b, 314 measurements) was found to favor the direct angle measurement for variance between individuals and errors in calculation, the pooled standard deviation was 6.249, and the standard error was 1.105 for a test statistic of t(30) = 3.11, (*p* < 0.005). The number of measurements from each method outside one standard deviation was totaled for right and left tibial torsion in all four cases and ranked from low to high in a Mann–Whitney U test. The direct method (R = 169.5) produced less measurements outside of one standard deviation p than compared to the horizontal line method (R = 358.5); the test statistic for the directional measurement was used, U_0.01,16,16_ = 169.5, *p* < 0.025 Participants who used the horizontal line method were more likely to measure outside of one standard deviation than the direct method *p* < 0.025.

## 4. Discussion

The purpose of this trial was to test how well orthopedic practitioners were able to measure axial alignment and malalignment using an online digital planning tool. There were three interventions (1) an online course which explained axial alignment via a lecture and discussion group format, (2) an online instruction on how to use a new digital education platform (Bonesetter app), and (3) four exercises which each test nuances in axial measurement. The simple fact that participants from around the globe tuned in for this 10-weeklong course with average live attendance of 170–230 participants proved that there was interest in this type of course and platform during the COVID-19 pandemic. Furthermore, it showed that we were able to use readily available meeting apps to accomplish this. We found however that only a proportion of the participants actually attempted the exercises that we proposed (case 1 = 80, case 2 = 61, case 3 = 55, case 4 = 42 participants). The effectiveness of the next two interventions were measured by the exercises. The first exercise was simply a patient with normal alignment parameters. This tested the participants ability to use the app correctly. It turned out that the participants were able to measure angles but many of them (35%) did not complete the full exercise. Case 2 was a test of understanding anteversion vs. retroversion and 60% of the participants performed the exercise correctly with 30% just not completing it and mostly because they did not do both sides in order to compare. Six percent added or subtracted the numbers incorrectly for the retroversion. Case 3 was to identify a large external rotation deformity which was both in the femur and the tibia and the participants had to figure that out and how much difference there was from side to side and where it was located. Even though most participants (84%) completed the exercise correctly, for the first time the errors were in calculation and measurement. Case 4 was performed correctly by 76% of participants and was again challenging to figure out when to add or subtract the numbers exclusively due to femoral anteversion or retroversion.

When performing the measurements, a portion of participants followed Dr. Bernstein’s instructions with the horizontal line method while others measured the angles directly. Overall, measuring angles directly produced smaller standard deviation per case than by comparing to the horizontal (t_1,30_ = 3.019, 0.0025 < *p* < 0.005). Direct measurement also had less incidents outside of one standard deviation from the example or mean (U_1,16,16_ = 206.5, *p* < 0.025), meaning that they obtained values closer to the central tendency than those using the horizontal line. Measuring the angles directly eliminated the need to manually calculate retroversion or anteversion, which was one of the largest sources of error (case 1 = 2.5%, case 2 = 1.6%, case 3 = 7.70%, case 4 = 9.50). Participants were able to measure malalignment accurately using both methods with an overall standard deviation of ±4.26 degrees.

None of the participants had used the Bonesetter app previously and >90% of the participants were able to perform measurements properly, and had thus effectively learned how to use the app to make digital measurements.

The Bonesetter app and other angle measurement tools can be used to measure the exact amount of rotational correction needed to fix malrotation. A unique feature within the Bonesetter app allows educators to view data saved by participants that can be analyzed remotely for accuracy in real time and be used to give direct feedback to each participant. This study shows that online webinars can be an effective tool to educate a large population in using the Bonesetter app to measure rotational deformities of the lower extremities. With this online course, surgeons can learn to use the Bonesetter app in an online setting to minimize travel and conference costs, and educators can see their work remotely in real time. The application can be used in addition to clinical gait observations to quantify the rotational corrections needed in an osteotomy procedure and leads to a greater confidence level for the surgeon, as well as a higher patient satisfaction rating. A unique feature within the app allows any user to upload cases and view data and other measurements from others and can streamline second opinions to help residents plan surgeries with attending supervision.

This study has several limitations. The application, and analysis, cannot determine the level of experience of participants. Many data sets were incomplete because online platforms have no way of enforcing full participation and many participants either stopped submitting data or only completed half the measurements for each case. Additionally, there is currently no standard for the location of the CT slice measuring the rotation of the acetabulofemoral joint. The absence of landmarks on the femoral neck made it more difficult for participants to identify the precise center of the femoral neck. The CT scans used to measure the tibial version in the online course deviated from the standard practice of using the proximal tibial condyles [17], and instead compared the bimalleolar axis to the femoral condyles [9]. The difference between the location of measurement used is likely insignificant because of all the participants’ abilities to accurately measure the distal femur vs. the more difficult proximal tibia.

There are no prior studies that have examined the effectiveness of an online course teaching a skill and then trying to validate the exercise with an online app. The COVID-19 epidemic forced us to retreat from live courses, but the technology and innovations developed will change our lives forever.

In future courses instructors should: (1) provide clear CT images with standardized measurement locations, (2) explain that the angle tools can be used to directly measure femoral version and tibial torsion by drawing the angles with respect to the condylar axis instead of with respect to the horizontal, and (3) clearly explain the landmarks of the femoral neck and bimalleolar axes to improve accuracy. Alternatively, the instructor of the course could explain retroversion and anteversion, internal and external rotation with greater care, as this was a common mistake within the data.

## 5. Conclusions

The study used the unique features of the Bonesetter app to evaluate the ability of orthopedic practitioners to measure axial rotation after receiving online instruction. In most cases, the average response for femoral version or tibial torsion was less than one standard deviation from the example provided by the instructor. The largest sources of error were not completing both the right and left sides of each case and confusing when to add or subtract angles (anteversion or retroversion) or calculation errors. Measuring the angles directly instead of with a horizontal line using the app improved accuracy and reduced errors. Future studies may further demonstrate the reliability of online training courses for the Bonesetter app with less experienced participants.

## Figures and Tables

**Figure 1 medicina-58-01079-f001:**
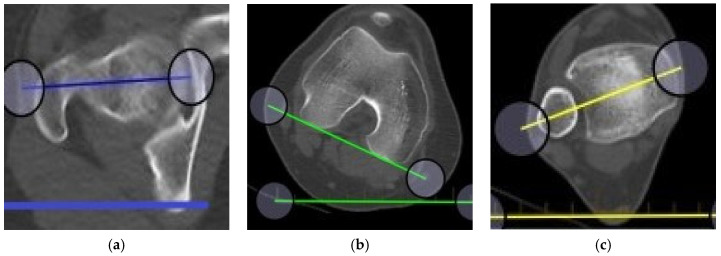
Measurement technique taught in the course: (**a**) right femoral head/neck axis; (**b**) right posterior condylar axis; (**c**) right bimalleolar axis.

**Figure 2 medicina-58-01079-f002:**
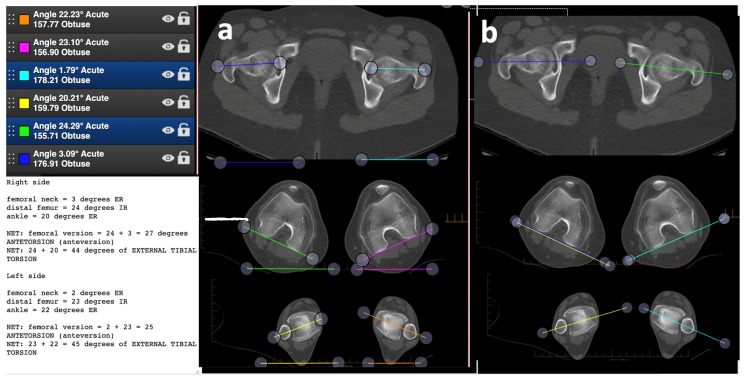
Examples of correctly measured angles on case 1. (**a**) Shows the angle lines drawn in the accepted example case. The top lines measure the femoral neck angle, the middle measures the posterior condylar angle, and the bottom measures the bimalleolar angle with respect to the horizontal; (**b**) this is an example of a variation in the measurement technique. Instead of comparing the angles to the horizontal, one can measure the femoral neck and bimalleolar angles with respect to the posterior condylar axis of the femur. This method eliminates addition and subtraction for femoral version and tibial torsion, but one must still determine if there is anteversion, retroversion, internal, or external tibial torsion.

**Figure 3 medicina-58-01079-f003:**
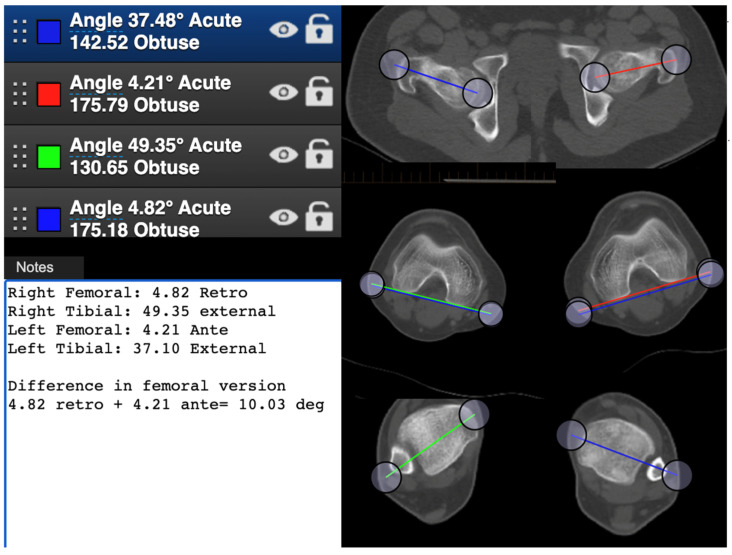
CT image provided for case 2; showing correct measurements, identification of femoral anteversion and retroversion, as well a calculation to establish difference in femoral version.

**Figure 4 medicina-58-01079-f004:**
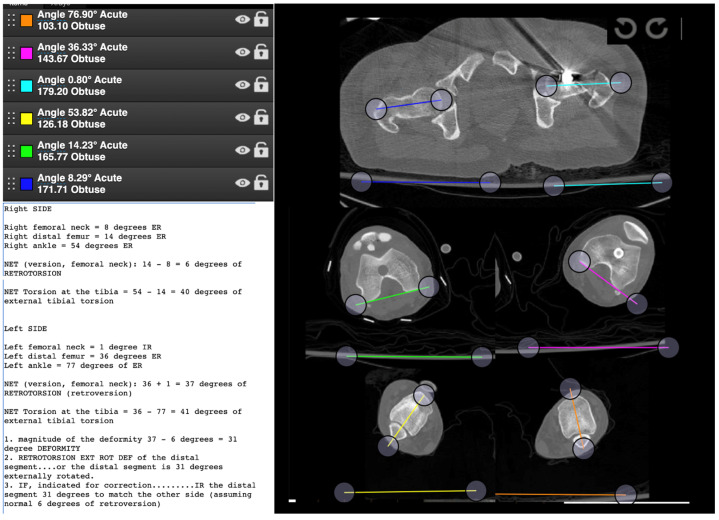
CT image provided during the case study.

**Table 1 medicina-58-01079-t001:** Summarizes reasons for error in each case.

Reason for Error	Case 1	Case 2	Case 3	Case 4
Case completed correctly	56%	61%	84%	76%
Participants only completed part of the measurements	35%	30%	4%	2%
Logged on but did not complete any data	1%	1%	0%	5%
Completed measurements but measured angles incorrectly	1.25%	6.60%	5.50%	7%
Completed measurements but made calculation errors	2.50%	1.60%	7.70%	9.50%

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
