# Peer review of "Online Instruction to Measure Axial Alignment with the Bonesetter App"

_medicina, 2022, doi:10.3390/medicina58081079_

Round 1
Reviewer 1 Report
1) The Bonesetter application attempts to provide the technology needed to measure rotational disorders in patients using computed tomography, but the use of this application is easily replicated with free systems. What would be the benefit?
2) What was the benefit of offering an online course on online planning and measurement during the pandemic?
3) The orthopedic surgeons carry out a daily planning with free digital tools, otherwise it would be difficult to carry out this type of surgery with a correct planning
I think you should explain these points in depth, so that the article can be publishable.
Reviewer 2 Report
The manuscript describes an important subject in surgical subjects. Here especially for the department of orthopaedics based on a developed app.
The manuscript is well written, but I recommend the following improvements:
Change of title:
Short and concise, and focus on the essential content.
Abstract:
Shortening, presenting results in a more differentiated way.
Please be more critical in the conclusion, as I would not consider the statement of the last sentence as generally valid, since too few have tried this app or this one.
It should be stated in more general terms that the implementation has been successful and that this can be used to support the future.
Introduction:
Shorter, more concisely present essential contents. Improve the transition to COVID.
and why was this app developed in the first place or what is the point of it? This should be further elaborated.
In addition, it should be pointed out that such digital applications are on the rise especially in orthopedics and are developing a field of their own.
Methodology section:
partly linguistically revised, otherwise well written.
Result section:
Very long, please shorten.
With the illustrations, make sure that the images have the same format sizes, often a and d larger than the rest.
Discussion:
shorten, revise the language and please elaborate on the limitations of the study.
It should also be pointed out how this app can be used in the future.
Reviewer 3 Report
1. I would recommend Authors could share the link of software/APPs access.
2. Please describe CT parameter requirements for this study.
3. For how long participants were taught to get a reliable duplicate?
4. Limitations of this study should be included.
Round 2
Reviewer 1 Report
Your work is originally and interesting but need some minor revision.
1)The abstract is well‐written,concise and clear
2)In the introduction hypothesis is clearly presented and is supported by text.
3)Materials and methods is clear
4) The results appear in order.
However, I do not find this metology innovative, already applied daily in surgical planning.
